# The Features of Potassium Dynamics in ‘Soil–Plant’ System of Sour Cherry Orchard

**DOI:** 10.3390/plants12173131

**Published:** 2023-08-31

**Authors:** Tatyana Roeva, Elena Leonicheva, Larisa Leonteva, Oksana Vetrova, Margarita Makarkina

**Affiliations:** Russian Research Institute of Fruit Crop Breeding (VNIISPK), Zhilina 302530, Orel Region, Russia; leonicheva@orel.vniispk.ru (E.L.); leonteva@orel.vniispk.ru (L.L.); vetrova@orel.vniispk.ru (O.V.); makarkina@orel.vniispk.ru (M.M.)

**Keywords:** sour cherry (*Prunus cerasus* L.), haplic luvisol, nitrogen and potash fertilizers, exchangeable, water-soluble and non-exchangeable potassium, leaf and fruit potassium content, fruit yield

## Abstract

This research aimed to study interannual and seasonal dynamics of different potassium compounds in orchard soil and the potassium status of sour cherry trees affected by the application of nitrogen and potash fertilizers. Afield experiment was started in 2017 at an orchard located in the forest-steppe zone of the Central Russian upland. Urea and potassium sulfate were applied to the soil once a year in early spring with rates from N30K40 to N120K160 kg/ha. The content of exchangeable and water-soluble potassium compounds was determined in soil samples five times throughout the growing season from May to September 2018–2020. The content of non-exchangeable potassium was determined twice, in 2017 and 2020. The interannual and seasonal dynamics of plant-available potash in unfertilized soil depended on the weather patterns and the uptake of potassium by trees. In the unfertilized plots, the first signs of potassium nutrition insufficiency appeared, such as low leaf and fruit potassium status and a decrease in the non-exchangeable potassium reserves in the20–40 cm soil layer. The annual fertilization led to the gradual accumulation of exchangeable potassium in the root zone. The accumulation was accelerated with increasing rates. When the exchangeable potassium level in the topsoil reached 200 mg/kg, the intensification of both the seasonal fluctuations in potash content and the potash leaching into the depths of the soil occurred in all treatments. In the conditions of our experiment, one-time treatments with superfluous potassium rates (over 80 kg/ha) did not provide an enlarged stock of plant-available potash in the soil but caused unreasonable losses of it due to leaching. An increase in fertilizer rates was not essential for normal metabolic processes and did not manifest itself as an increase in potassium content in leaves and fruits or as an increase in yield.

## 1. Introduction

Sour cherry trees uptake considerable amounts of nutrients from the soil, and the uptake level of potassium is the second highest, after nitrogen. Annual potash intake by 8–10-year-old sour cherry trees is 27.9 kg/ha [1].

This element is an essential nutrient for sour cherry productivity and fruit quality because of its important role in many physiological processes. Potassium is not part of the organic macromolecules but is present in plant cells in its ionic form (K^+^), which serves as the most important osmotic compound and has an effect on turgor-driven processes like stomatal movement [2,3,4]. Another essential function of potash is connected to its participation in photosynthetic processes. The element’s deficiency is associated with a reduction in the photosynthetic capacity of CO_2_ assimilation and anatomical alterations in leaf structure [5]. Also, potassium contributes to the long-distance phloem transport of photoassimilates. This process is especially important for the fruit trees, which require the translocation of sucrose from photosynthetically active organs (leaves) into sink ones (fruits) [2]. A positive impact of fertilization with potassium on fruit tree productivity has been repeatedly reported [6,7,8,9].

On the other hand, excessive rates of potash fertilizers decrease Ca, Mg and Mn uptake by cherry trees and lead to degradation in fruit quality [10,11]. Both a deficit and abundance of potassium can result in the imbalance of physiological processes and decrease plant productivity [2,12,13].

Potassium is removed from stone fruit orchards with yield and often with pruned shoots, especially in intensive horticulture. Annual removal impoverishes the soil supply of plant-available potassium compounds. The long-term growing of fruit trees without fertilization led to a significant decrease in exchangeable potassium reserves in orchard soil over 6–8 years [14,15]. However, sometimes the rates of potash-containing fertilizers applied in orchards significantly exceed the plant needs for this nutrient [16]. Such excessive fertilization results in the decrease in fertilizers’ agronomic efficiency [17]. Thus, for sustainable stone fruit orchard production it is necessary to apply potassium fertilization with rates according to tree needs and to ensure profits for fruit growers. These optimal rates may vary significantly depending on soil–climatic conditions and demands of fruit crops and cultivars.

To determine the optimal potassium rates for orchards planted in different soil types, it is necessary to combine various methods of soil nutrient investigation with the chemical analysis of plant tissues. The availability of soil potassium for plants is linked to the inter-relation of different potassium compounds being in dynamic equilibrium (exchangeable, non-exchangeable and water-soluble) [18]. The study of these inter-relations enables a complex view of the potassium issue in the orchard agroecosystem.

In the Russian Federation, the assessment of the potassium supply in agricultural soils is based on the determination of exchangeable potassium. These compounds are extracted from the soils with the help of 0.2 mol/dm^3^ HCl (Kirsanov method), 0.5 mol/dm^3^ CH_3_COOH (Chirikov method) or a 1% solution of (NH_4_)_2_CO_3_ (Machigin method), which are recommended for acidic, neutral and calcareous soils, respectively [19].

Water-soluble potassium is the most mobile part of exchangeable potassium [18]. It is most accessible to roots and moves easily in the soil profile. The amount of the element extracted by water characterizes the current level of plant potassium nutrition. The content of water-soluble potassium reduced by plant uptake usually replenishes quickly from the reserves of exchangeable compounds with sufficient soil humidity. Thus, this indicator reveals the ability of the soil to desorb potassium ions into the soil solution.

The non-exchangeable potassium, in turn, is a reserve for the replenishment of more mobile bioavailable compounds. Plant roots release H^+^ into the rhizosphere [20], which results in the release of non-exchangeable potassium from clay minerals into the soil-adsorbing complex [21]. The application of NH_4_^+^-containing fertilizers also leads to alterations in the ratio of potassium compounds with unequal availability for plants [22]. In long-term field experiments, the non-exchangeable potassium content in topsoil and in subsoil is a sensitive indicator revealing the early stages of soil degradation [23].

Stone fruit orchards grow on the same plot for a long period, and potassium uptake by trees is continuous, so the potash status of the orchard soil changes gradually. The floor management system and irrigation and nutritional management also affect the inter-relation of potassium compounds in orchard soil from year to year.

Fertilization with potassium is often ineffective in the first years after planting an orchard if the soil originally had heavy texture and favorable agrochemical properties [24,25,26]. However, with increasing tree age, the efficiency of potash fertilizers usually increases, and 10–20-year-old trees are more responsive [8,27,28,29]. Therefore, the need for potassium in fruit trees increases as they grow and enter the fruiting stage, and the rates of potash fertilizers applied in the orchard should be increased according to the needs of the plants. To determine the optimal period to begin fertilizer application, long-term field experiments should be carried out in orchards growing in various soil–climatic conditions.

Currently, the potassium regime of orchards is poorly studied, and long-term data are collected mostly for pome crops. Sergeeva et al. [30] ascertained that without fertilization the content of exchangeable potassium in chernozem gradually decreased by half over 10 years after planting an apple orchard. In the next 10 years, this indicator remained consistently low, which measured up to a new equilibrium ratio of different potassium compounds in the soil-absorbing complex. Kuzin et al. [31], within two vegetation periods, investigated the seasonal dynamics of exchangeable potassium in the meadow-chernozem soil of a high-density apple orchard and recorded similar dynamics in two experimental plots planted with different cultivars. In this experiment, the potassium content varied depending on the soil moisture and fertilizer treatment, but every year the lowest potassium level was in August and September (intensive growth and ripening of apples).

The interannual dynamics of plant-available potassium in the 0–20 cm layer of Humic Cambisol was studied in a pear orchard during 2010–2017 [24,27]. In the topsoil (0–10 cm) of unfertilized plots, the potassium content decreased sharply (about 40%) in the second year of the experiment, and in subsequent years the index was relatively stable. In the subsoil (10–20 cm), the decreasing of available potassium was more gradual: it decreased by three times in five years.

Currently, the potassium regime of stone fruit orchards is much less studied than that of pome ones. Nutritional management for stone fruit crops is often based on the assessment of fertilizers’ effect on fruit yield and quality with no consideration of soil properties. The complex view of the potassium issue in stone fruit orchards regarding both the plant diagnosis of potash nutrition and the study of soil potassium ‘behavior’ in specific soil and climatic conditions may be useful for the elaboration of precise nutritional management for specific crops.

The purpose of this research was to study the interannual and seasonal dynamics of different potassium compounds in orchard soil and the potassium status of sour cherry trees affected by the application of nitrogen and potash fertilizers. The results of this research might be useful for adjusting the rates and timing of fertilization with potassium in sour cherry orchards growing in conditions of the East European Plain.

## 2. Results and Discussion

### 2.1. The Dynamics of Different Potassium Compounds in Soil

#### 2.1.1. Interannual and Seasonal Dynamics of Exchangeable Potassium

The distribution of potassium compounds in the ‘soil minerals–soil colloids–soil solution’ system and the equilibrium ratios between these compounds determine the potash status of the soil and its ability to obtain the needs of plants for this nutrient. The exchangeable potassium is the part of the soil potassium supply that is most important for the nutrition of agricultural crops due to its participation in establishing the equilibrium concentration of potassium in the soil solution. The potassium status assessment of agricultural soils based on exchangeable potassium content is widely used in countries with varied soil–climatic conditions [18,23,27]. At the experimental orchard in this study, the exchangeable potassium content in the unfertilized soil varied from148.6 to 160.6, 83.3 to 97.0 and 57.9 to 68.8 mg/kg in the layers at 0–20, 20–40 and 40–60 cm depths, respectively (Table 1). The potassium level in the surface soil layer is classified as medium (100–200 mg/kg) according to the grading for fruit crops applied in the European part of Russia [32]. In the lower layers, the potassium level corresponded to the low range (<100 mg/kg).

The content of exchangeable potassium compounds in orchard soil primarily depended on weather conditions and fertilizer rates. The lowest potassium level was recorded in unfertilized plots. However, sour cherry trees growing without fertilization did not show a decrease in this indicator within 3 years (Table 1). Most likely, the studied soil was able to compensate for the reserves of exchangeable compounds with non-exchangeable potassium. The main parent rock materials of Haplic Luvisols in the Central Russian Upland are loess loams, which are abundant in hydrous micas. These 2:1 clay minerals are capable of both fixing and releasing potassium from the soil solution [33].

The strong fixation of potassium by soils is most active in the early years of fertilizer application. Over time, non-exchangeable fixation decreases due to the filling of the exchange capacity of the soil-absorbing complex, and the content of plant-available potassium compounds begins to increase [19]. With the regular use of potash fertilizers, the content of exchangeable potassium stabilizes at a certain level specific to a particular soil [34].

The accumulation rate of exchangeable potassium in the orchard soil depends on the rates of fertilizers. The level of exchangeable potassium in the surface soil layer already significantly increased (by 24–74%) compared with the control in the second year of fertilization (2018) at rates of N60K80 and higher. In 2020, all treatments led to a similar effect, and the level of exchangeable potassium reached a high range (>200 mg/kg). Similar increases in exchangeable potassium reserves in the soils of fruit orchards due to regular applications of potash fertilizers have been recorded in numerous investigations [15,24,27,35,36,37,38].

It is believed that potassium is mainly adsorbed in the upper layer of the soil and weakly migrates to the deeper layers. The potash leaching beyond the root layer can occur with the prolonged use of fertilizers and deep rainfall penetration into the soil [19]. During the fertigation of clay chernozems in the North Caucasus region of Russia, mainly lateral migration of potassium occurred, while the element moved slightly into the depths of the soil [39]. However, the vertical migration of potassium during fertigation was recorded in loamy soils of apple orchards in the central part of the East European Plain [40]. Prolonged soil application of potash fertilizers at rates of more than 150 kg/ha in irrigated orchards led to the removal of exchangeable potassium to a depth of more than 50 cm, especially with a large amount of irrigation water [41]. In rainfed pome orchards, vertical migration of potassium also occurs, and the depth of potassium removal depends on soil texture and humidification conditions [24,27,35,37,38].

Our results are consistent with the data from the above studies. Since 2018 at the experimental sour cherry orchard, the application rates of N60K80 and higher were sufficient to significantly increase exchangeable potassium at a depth of 20–40 cm. After the four years of fertilizer application (in 2020), a significant increase in this index was also observed in the 40–60 cm layer when rates of N60K80 and higher were applied (Table 1).

The main factors affecting the seasonal fluctuations in soil exchangeable potassium were the weather patterns and the uptake of potassium by trees. In the first half of the 2018 growing season, the level of exchangeable potassium in the topsoil fluctuated slightly due to drought from late May to mid-July. Rainless conditions reduce the availability of soil potassium for plants because it diffuses into roots via the films of water surrounding soil particles [42]. In the dry season of 2018, the sour cherry trees could remobilize necessary potassium from storage reserves within the plant. The replenishment of the internal potassium supply by trees became possible due to external nutrient uptake after heavy precipitation (119 mm) from July 13 to 25. Thus, in August 2018, the content of exchangeable potassium decreased in the 0–20 soil layer regardless of experimental treatments (Table 2).

In the soil–climatic conditions of the East European Plain, the essential decrease in exchangeable potassium in the chernozem soil of apple orchards occurs in the period of apple ripening (August) [31]. The sour cherry fruiting period is relatively short compared to other temperate fruit crops, and it takes about 2 months from flowering to ripening. Therefore, the potassium uptake by sour cherry trees may differ in timing. Thus, in 2019, we observed in the 0–20 cm soil layer the diminution of exchangeable potassium in June and July both in fertilized plots and in the control ones (Table 3).

A characteristic feature of weather conditions in the 2020 growing season was regular prolonged rainfalls from May to July. Soil water content in the root zone during this period was over 20%. As a result, we observed leaching of soil potassium to the deeper soil layers (Table 4) and lower leaf and fruit potassium status To maintain potassium homeostasis, sour cherry trees required intensive uptake of the element from the soil, which led to an essential decrease in exchangeable potassium in the 0–20, 20–40 and 40–60 cm soil layers in August and September (Table 4).

The content of exchangeable potassium compounds in deeper soil layers (20–40 and 40–60 cm) fluctuated during the vegetation period less sharply than in the topsoil. The values of the indicators temporarily increased after intense precipitation mainly in the soil of plots fertilized at rates of N60K80 or higher. In more wet months (especially in July 2020), the index in fertilized sites was 1.5–2 times higher than the control due to potassium leaching from the upper layer.

#### 2.1.2. Interannual and Seasonal Dynamics of Water-Soluble Potassium

Water-soluble potassium compounds are the main resource of plant nutrition. The uptake of potassium by plants reduces its concentration in the soil solution, but then the potassium level rapidly replenishes via cations released from the exchangeable positions. The balance between water-soluble and exchangeable forms of potassium depends on soil properties such as pH, CEC, and clay mineralogy. It could be influenced by the alteration of ions in the soil solution and the total concentration of soluble anions [43]. In addition, the distribution of water-soluble potassium in the soil profile is due to bioaccumulative processes and features of eluvial–illuvial differentiations of soil [44].

In the different soil groups of Central Europe, the share of water-soluble potassium in the total potassium variesfrom0.7 to 5% depending on soil texture [43]. The same proportion between the water-soluble and total potassium was observed in the arable calcareous soil of China, and this proportion significantly depended on the application of both potash-containing and NH_4_^+^-containing fertilizers [22]. In agricultural soils of Pakistan, the share of water-soluble potassium in its total content varied from 0.03 to 0.05%, and the water-soluble potassium content was 25–27 times lower than that of exchangeable potassium [45].

The level of water-soluble potassium in the unfertilized soil of the experimental orchard varied from 10 to 20 mg/kg in the 0–20 cm soil layer and from 6 to 12 mg/kg in deeper layers. The fertilization with potassium sulfate at rates of K40–K160 led to an increase in the water-soluble potassium content. However, the augmentation of this index was not proportional to the increase in fertilizer rates. Unlike the exchangeable potassium, the water-soluble potassium did not accumulate in the orchard soil with the regular use of potash fertilizers. Habib et al. [45] recorded similar results for clay loamy soil treated with potassium sulfate for 26 years. The content of water-soluble potassium in this soil was only 1.5–2 mg/kg higher than in unfertilized soil.

The application of the lowest fertilizer rate of N30K40 contributed to the statistically significant increase in water-soluble potassium only once, in 2019, in the 0–20 cm soil layer. A similar effect of rates of N60K80 and above was more stable and observed in both topsoil and subsoil layers (Table 5). The largest dose of fertilizer, N120K160, did not provide the highest level of water-soluble potassium. With the doubling of the fertilizer rate (from N60K80 to N120K160), the close values of the indicator were observed annually.

The seasonal fluctuations in water-soluble potassium in soil with no potassium application are generally insignificant compared to the dynamics of exchangeable compounds [46]. In our experiment, the differences between the lowest and the highest levels of water-soluble potash in the unfertilized soil were not higher than 10 mg/kg (Table 6, Table 7 and Table 8). The treatments with potash fertilizers intensified fluctuations in the 0–20 cm soil layer, especially when rates of N60K80 and higher were applied (Table 6, Table 7 and Table 8).

Similar to the seasonal dynamics of exchangeable potassium, the dynamics of water-soluble depended on the weather features of the current vegetation period. Due to drought in the first half of the 2018 growing season, the content of water-soluble potassium in the 0–20 cm layer fluctuated slightly in all experimental treatments except forN90K120 (Table 6). The decrease in water-soluble potassium during fruit ripening occurred in all experimental treatments only in 2019 (Table 7).

By observing the seasonal dynamics of water-soluble potassium in deeper soil layers, we detected that the water-soluble potassium content at the depths of 20–40 and 40–60 cm increased periodically after rainfall, but only in the treatments with N60K80 and higher (Table 6, Table 7 and Table 8). During the 2020 rainiest vegetation period, the level of water-soluble potassium at a depth of 20–60 cm in all experimental treatments was mostly low, which is probably due to the quick potash removal outside the studied soil layer (Table 8).

The N90K120 treatment provided the highest values of both water-soluble and exchangeable potassium over the three growing seasons (2018–2020) (Table 1, Table 2, Table 3, Table 4, Table 5, Table 6, Table 7 and Table 8). Seasonal fluctuations in water-soluble potassium were also most intense in the soil fertilized with N90K120 (Table 6, Table 7 and Table 8).

Suppose that the treatment with N90K120 in the early years of fertilization already provided the saturation of the soil-absorbing complex with potassium. The indicators of this maximum saturation for the studied soil are the indices of water-soluble and exchangeable potassium close to 30 mg/kg and 200 mg/kg, respectively. An excess of these levels increases potassium mobility in soil, and seasonal fluctuations are also intensified.

It should be considered that potassium sulfate and urea in our experiment were applied at the same time at simultaneously increasing rates. Therefore, the additional cations of NH_4_^+^ regularly come into the absorbing complex and compete with K^+^ for exchange positions. Thus, with the application of the maximum rate of N120K160, potassium may be less retained by the soil and more easily move beyond the root layer. As a result, the levels of water-soluble and exchangeable potassium were often lower with the N120K160 treatment than with the treatments with smaller rates (Table 2, Table 3, Table 4, Table 6, Table 7 and Table 8).

#### 2.1.3. The Alteration of Non-Exchangeable Potassium Content Affected by 4-Year Fertilizer Application

The determination of non-exchangeable potassium was carried out twice during the experiment—at the end of September 2017 (1st year of fertilization) and 2020 (4th year of fertilization). The content of non-exchangeable potassium in the topsoil of the unfertilized plots was ten times higher than the exchangeable and varied from1249 to 1549 mg/kg. In deeper soil layers, this index varied from1085 to 1424 and 1003 to 1315 mg/kg at the depths of 20–40 and 40–60 cm, respectively.

The application of potash fertilizers for four years did not affect the content of non-exchangeable potassium in the 0–20 and 40–60 cm layers. However, without fertilizers at a depth of 20–40 cm, the indicator value in 2020 (1117 mg/kg) was 1.3 times lower than in 2017 (1424 mg/kg) (Figure 1).

Consequently, in this soil layer, the displacement of the equilibrium dynamic between different potassium compounds occurred as the effect of root uptake of the element. As mentioned above, the stable level of exchangeable potassium in the unfertilized soil could be maintained by releasing non-exchangeable potassium from minerals into the soil-adsorbing complex. At the same time, the level of non-exchangeable potassium in the 20–40 cm soil layer was stable when potash fertilizers were applied.

The absence of a significant increase in reserves of non-exchangeable potassium during the permanent application of fertilizers is possibly connected to the saturation of the soil-absorbing complex and boundaries of the clay minerals with other cations. The underlying rock of the loamy soil in the experimental orchard is dolomite limestone. Consequently, the main saturating cations in soil are Ca^2+^ and Mg^2+^ Moreover, an additional amount of ammonium cations, formed as a result of the transformation of urea, entered the soil annually. The competition between K+ and NH_4_+ cations in the fixation and release processes in 2:1 clay minerals is well known [47]. Balík et al. also reported the insignificant impact of long-time (21 years) fertilization on the level of non-exchangeable potassium in Haplic Luvisol [18].

### 2.2. Plant Potassium Status, Fruit Yield and Fruit Quality Parameters

#### 2.2.1. The Dynamics of Leaf Potassium Content

Leaf analysis is a routine used to estimate the current nutritional status of fruit trees. The optimal range of potassium content in sour cherry leaves is 1.10–1.85% DW [48,49]. A similar level of leaf potassium of 1.08–1.63% DW was recorded by Rutkowski and Lysiak [50] in conditions of Western Poland. In our experiment, the leaf potassium status of sour cherry trees in the course of three vegetation periods was significantly lower and varied from 0.49 to 1.09% DW.

Usually, leaves collected much earlier tend to contain higher potassium concentrations [48]. We observed a similar trend in 2019 and 2020, but in 2018, leaf potassium status was stable from June to August in all treatments (Figure 2).

The vegetation period of 2018 differed from the next two in contrasting humidification conditions and the low productivity of the sour cherry trees (not over 5 kg/tree). Therefore, the weather conditions and fruit load were the main factors affecting the dynamics of leaf potassium.

The dependence of leaf potassium status on plant productivity has been reported for pome crops [24,31,51], and Rutkowski et al. [52] reported a significant effect of meteorological conditions on the potassium content in sour cherry leaves. In our experiment, the influence of weather conditions was especially noticeable in 2020, when prolonged rainfall led to potassium leaching from leaves (Figure 2). Nutritional losses affected by heavy precipitation have been demonstrated for European broadleaf trees in conditions of a temperate continental climate [53].

In the orchards, the inter-relation between soil and leaf potassium is often ambiguous. On the one hand, the leaf potassium status is higher if sour cherry trees are planted in soils rich in plant-available potassium [54], and fertilization with potash optimizes its level in leaves if trees are growing in soil with a deficit of this element [11,55,56]. On the other hand, the increase in fertilizer rates by several times often does not cause a proportional increase in leaf potassium status [11,27].

In the studied orchard, the application of potash fertilizers slightly affected leaf potassium content. The values of this index did not increase above 1.15–1.2 times in 2018 and 2019 (Figure 2). Supposedly, the initial potassium level in orchard soil was satisfactory to meet the trees’ needs for this nutrient in the first years after planting. In the course of increasing yield and the enhancement of tree biomass, sour cherry needs for potassium increased. In 2020, when average yield was 8.4 kg/tree, leaf potassium status was the lowest over the entire period of observation. The high yield and potash losses due to leaching provided the most visible effect of fertilizer treatments on leaf potassium status in 2020. The potash concentration in leaves of fertilized trees this year was 1.2–1.4 times higher than that of the control. However, the increase in fertilizer rates from K40 to K160 did not cause an additional increase in leaf potassium status. Similar data were obtained by Yener and Altuntaş [11], who did not reveal an increase in the content of potassium in sweet cherry leaves when potassium rates were increased from 100 to 600 g/tree.

#### 2.2.2. Fruit Yield, Fruit Potassium and Fruit Quality Parameters

The essential role of potassium in the processes of the long-distance transport of sugars and water is well known [2]. Thus, the optimization of potash nutrition leads to the increase in fruit weight and fruit size for various fruit crops such as sweet cherry [11,57], pear [29] and apple [28]. However, a positive influence of mineral fertilizer on the size and weight of fruits is unstable, whereas the effects of meteorological conditions, yield load and varietal features are more noticeable [58].

In our experiment, the fruit weight varied from4.38 to 5.31 g over three years and was not affected by fertilizer treatments and weather conditions (Table 9).

Unlike the fruit weight, the productivity of the sour cherry trees responded to fertilizer application, but only in 2020 (Table 9). This year, the treatments at rates of N60K80 and higher promoted 1.5–1.7 times higher yields than those of the control. However, the doubling of fertilizer rates from N60K80 to N120K160 did not cause an additional increase in productivity.

As it was mentioned above, the initial potassium level in the orchard soil was satisfactory to meet the young trees’ needs. Also, earlier we found out that loamy Haplic Luvisols in the climatic conditions of the Central Russian Upland can provide a favorable level of nitrogen nutrition for sour cherry trees in the first years of fruiting without the application of nitrogen fertilizers [59]. With the increasing age of the trees, the nutritional requirements may increase, enhancing fertilizer efficiency. A positive effect of fertilization was recorded for various fruit crops mostly when a tree’s age was above 10 years [8,27,28,29,60].

The potential productivity of mature cv.‘Turgenevka’ sour cherry trees is about 30 kg/tree [61]. At the fifth and sixth years after planting (2019 and 2020, respectively), only one-third of the maximum productivity was obtained (Table 9). Nevertheless, in 2020, the first signs of potassium nutrition insufficiency, such as low leaf and fruit potassium status and a decrease in non-exchangeable potassium reserves in the 20–40 cm soil layer, appeared in the unfertilized plots. Additional potassium losses during this vegetation period might be due to leaching from the above-ground part of the trees. All these reasons together can be associated with the improved plant potassium status and increased yield observed in the 2020 potash fertilizer treatments in the study orchard.

It should be considered that fertilization with N and K was performed simultaneously in our experiment. The study of the nitrogen regime in the experimental orchard showed that the soil provided a sufficient level of nitrogen nutrition for young and fruiting sour cherry trees with the absence of nitrogen fertilizers [59]. However, as shown above (Figure 2), the potash nutrition of the control trees was unsatisfactory. Consequently, the positive contribution of additional potassium nutrition to the increase in yield is beyond doubt.

The potassium status of sour cherry fruits often does not depend on the level of soil potassium and fertilizer rates [62,63], especially if the orchard soil is rich in plant-available potassium. Nevertheless, the value of this indicator may depend on the productivity of trees [64]. In our experiment, the fruit potassium status was not affected by yield, while the influence of weather conditions was significant. The average yield of the studied trees in 2019 and 2020 was similar (8.34 and 8.39 kg/tree, respectively), but the fruit potassium content was significantly lower in 2020 due to the leaching of potash by prolonged periods of precipitation (Table 10).

In our experiment, both leaf and fruit potassium status were susceptible to fertilizer application. However, the relation between fruit potassium and fertilizer rates was unclear. With the application of the maximum rate of N120K160, the fruit potassium status did not differ from the control for three years. Simultaneously, lower rates promoted a significant increase in fruit potassium, but indicator values of unequal treatments did not differ from each other (Table 10). Presumably, the lowest potassium rate of 40 kg/ha was sufficient to stabilize the equilibrium between soil potassium compounds and to maintain the stable potassium regime of young trees. An increase in fertilizer rates was not essential for normal metabolic processes and did not manifest itself as an increase in potassium content in leaves and fruits.

Long-term (1990–2017) investigations of fruit quality traits indicated that sour cherry cultivars growing in the soil–climatic conditions of the Orel region had the following parameters: TSS varying from 10.8 to −19.8%, TA—0.73 to −2.62% and TS—8.36 to −14.26% [65]. Compared to these data, the TSS content in the sour cherries of our experiment was relatively high, especially in 2020 (Table 10). Due to low temperatures in May 2020 and heavy precipitation from May to July 2020, the period of fruit growth and ripening was delayed more than usual, and fruits were harvested in the second half of July. Higher air temperatures and sufficient rainfall at the late stage of fruit ripening promoted the accumulation of substances with high nutritional value, such as sugars and organic acids. The essential effect of weather features on TSS and TA was also recorded for apples [66] and sweet cherries [67].

In our experiment, the TSS content in sour cherries did not depend on fertilization during all three years of the experiment, which is consistent with the results of Vetrova et al. [68] for apples in similar soil–climatic conditions.

It has been recorded that potash fertilizer increases the sugar content in various kinds of fruits like citrus [69], apples [70], pears [29] and tomatoes [71]. In a study by Wu et al. [71], the potassium rate of 0.2 g/kg increased the sugar content compared to the control, but doubling and tripling of the rates did not lead to proportional increases in the sugar content in tomatoes.

In our experiment, the response of the TS index to the fertilizer treatments was dissimilar in the 2019 and 2020 vegetation periods. In 2019, the total sugars in sour cherries tended to decrease after fertilization, but in 2020, a significant increase in TS was observed after applying N60K80 (Table 10).

Potash fertilizer application decreased fruit acidity by decreasing the malic acid concentration in apples [70]. However, in a study by Yener and Altuntas [11], the titratable acidity in sweet cherry fruits increased with potassium rates. In our experiment, the TA index tended to increase due to the effect of fertilization only in 2020, when the TA level was maximum, but a significant increase in TA was recorded only when applying N90K120 (Table 10).

### 2.3. Relationships between Potassium Status Indices in ‘Soil–Plant’ System and Fruit Quality Parameters

Relationships between fruit quality parameters and potash concentration in the sour cherry leaves and fruits were estimated by a correlation analysis. All indices associated with the intensity of plant metabolic processes (TSS, TA and TS) correlated negatively with leaf potassium levels and fruit potassium levels (Table 11). This negative correlation may be caused primarily by the leaching of potassium from the above-ground part of trees, which occurred in 2020 simultaneously with the accumulation of sugars and other assimilates in the fruit. On the other hand, as it was shown above (Figure 2, Table 10), if the potassium content in plant tissues is sufficient for the normal course of metabolic processes, it does not increase proportionately to the increase in soil potassium supply. At the same time, the accumulation of organic substances and fruit growth continue and lead to the dilution of minerals in plant tissues. In our experiment, the dilution of potassium with fruit growth was confirmed by a negative correlation (r = −0.56; *p* < 0.05, *n* = 45) with fruit potassium status and fruit weight. Our results are inconsistent with those of Azizi et al. [54], who found no significant correlation between TSS and TA in fruit and potassium content in sweet cherry leaves.

Since potassium is required for most of the vital metabolic functions, it easily moves between plant organs and tissues [72,73]. This mobility complicates the identification of relationships between the indicators describing potassium status in the ‘soil–tree’ system. Plant nutritional status is determined in most studies related to the mineral nutrition of fruit crops. However, there are much fewer investigations comparing the results of soil and plant diagnostics. The correlation coefficients among the potash content in soil and leaves is low, as a rule, and does not definitely confirm the positive relationships between the indices [32,52,74,75]. In our experiment, the correlation between potassium content in sour cherry fruits and leaves was expectedly positive and high (r = 0.65; *p* < 0.01). At the same time, the relationships between fruit potassium status and the content of plant-available potassium in different soil depths were insignificant (Table 12). In turn, leaf potassium status correlated positively with the contents of exchangeable and water-soluble potassium compounds in the 20–40 cm soil layer (Table 12).

## 3. Materials and Methods

### 3.1. The Soil and Meteorological Conditions

The investigation of the relationship between soil potassium conditions and potassium content in fruits and leaves of sour cherry were carried out at the experimental site of the Russian Research Institute of Fruit Crop Breeding, located in the forest-steppe zone of the Central Russian upland (Orel region), Russia. The Orel region is located in a temperate continental climate zone at an altitude of 203 m above sea level. The absolute maximum in summer is +40 °C, the sum of temperatures above +10 °C is 2250 °C and the growing season lasts 175–185 days. The climate in the region is moderately continental with an average annual temperature of 5.5 °C and an average annual precipitation of 450–550 mm. Soil of experimental site is classified as loamy Haplic Luvisol (IUSS Working Group WRB, 2015) with favorable agrochemical characteristics which are presented in Table 13.

The temperature regime during the 2018–2020 vegetation period was close to the long-term average, and in some months, the temperature exceeded the average monthly level by 0.9–3.5 °C. However, May 2020 was cold, and was unfavorable for fruit setting (Table 14).

The total amount of precipitation and its regularity varied significantly in different years. Contrasting moisture conditions were a feature of the growing season in 2018: two drought periods were observed from late May to mid-July and in August, while 119 mm of precipitation fell from July 13 to 25. In 2019, the dry period lasted from late May to the third decade of June; in the following months, the periods of precipitation were more regular. In 2020, the greatest amount of precipitation (217 mm) fell from May to July (the period of fruit growth and ripening) (Table 15). In 2020, due to heavy rainfall, the fruits ripened 10 days later than usual.

### 3.2. Experimental Design and Treatments

We studied the interannual and seasonal dynamics of different potassium compounds in the soil and potassium status of sour cherry trees in the rainfed orchard planted in 2015 with an allocation scheme of 5 m × 3 m. In this orchard, the field experiment in studying mineral fertilizers’ efficiency started in 2017.

The soil of the experimental orchard initially had a high content of available phosphorus. Because of this feature, nitrogen and potassium fertilizers were chosen for the experiment to optimize the nutritional management of the studied crop. Organic and mineral fertilizers were not applied in the orchard before the start of the research.

The experimental treatments were as follows: 1. without fertilizers (control); 2. N30K40; 3. N60K80; 4. N90K120; 5. N120K160 kg/ha. The applied treatments based on the results of infrequent studies of sour cherry nutrition were performed in similar soil and climatic conditions. The rate of nitrogen and potassium in these investigations varied from60 to 180 kg/ha [76,77]. Bearing in mind the modern tendency to minimize the fertilizer rates in stone fruit orchards [78], we also applied the minimum rate of N30K40.

Fertilizers in the form of urea (46% N) and potassium sulfate (52% K_2_O) were applied annually in early spring (April) to a depth of 10–15 cm in a 2.2 m wide strip with the center in a row of trees. The experiment was carried out in three repetitions with a randomized arrangement of plots with 4 measurement trees in each plot. The soil management in the rows of trees consisted of herbicide treatments, but in the inter-rows, ploughing was conducted from 2015 to 2019 and the grass in the inter-rows was mowed from 2020 onwards.

The experiment was performed with sour cherry tree cv. ‘Turgenevka’ on the *Prunus mahaleb* rootstock. The ‘Turgenevka’ cultivar was chosen due to its frost and disease resistance. The fruits of this cultivar are popular with consumers for their large size (5 g) and dark red, juicy dense flesh.

### 3.3. Soil Sampling and Analysis

Soil samples were taken between trees in the subcronal zone at a distance of 1.0–1.2 m from the tree trunk at depths of 0–20, 20–40 and 40–60 cm. Sampling was carried out with the soil auger five times during the growing season from May to September 2018–2020. Mixed samples (about 500 g) were made of three point samples from each experimental plot of each repetition. Thus, every month we took 45 soil samples. The collected samples were dried at room temperature for a week and grounded and sieved through a 2 mm sieve.

The content of exchangeable and water-soluble potassium compounds was determined in these samples using the flame photometric method. Exchangeable potassium was extracted with 0.2 mol/dm^3^ HCl solution at a 1:5 soil:solution ratio. The suspension was shaken for 1 min on a horizontal shaker and was filtered for15 min after shaking [79]. The 1:5 soil:water ratio was used to extract water-soluble compounds. The suspension was shaken for 3 min on a horizontal shaker and was filtered for 5 min after shaking [79].

Soil sampling to determine non-exchangeable potassium was carried out twice, at the end of September 2017 (1st year of fertilization) and in 2020 (4th year of fertilization). Non-exchangeable potassium was extracted from the soil with10% HCl at a1:5 soil:solution ratio. The suspension was kept in the thermostat at 90 °C for1 h and then was filtered [79]. The non-exchangeable potassium of soil colloids and potassium fixed by soil from fertilizers were extracted by this method. The potassium concentration in extract was also measured with a flame photometer. The level of non-exchangeable potassium in soil was calculated as the difference between the value of potash in the extract with 10% HCl and the content of exchange compounds extracted with 0.2 mol/dm^3^ HCl.

### 3.4. Plant Sampling and Analysis

The leaf samples were taken 3 times during the growing season in June (30–40 Days After Full Bloom (DAFB), July (60–70 DAFB) and August (90–105 DAFB). From each plot of each repetition, 40 fully developed leaves were collected around the trees from the middle part of the annual shoots. Leaves were dried at room temperature for a week and then in a drying chamber for 6 h at 40 °C. Then, all plant material was homogenized. The dry leaf material was a shed in a muffle furnace at a temperature of 450 °C for 32 h. The ash was dissolved in 20% HCl. The resulting solution was diluted 10 times and then the potassium content was determined.

Mixed fruit samples (~1000 g) were taken from each plot of each repetition during harvest. The small portions of whole fruits (22–26 g each) were randomly picked from mixed samples and dried at a temperature of 70 °C in a drying chamber for about 96 h. Then, the analysis was performed similarly to that of the leaf samples.

The determination of potassium in ash solution and in soil extracts was performed with a Sherwood 410 flame photometer (Cambridge, Great Britain)

The mixed fruit samples were also subject to the evaluation of fruit quality traits. Total soluble solids (TSS%) were measured with a PAL-3 Digital Refractometer for Refractive Index and Brix (ATAGO, Tokyo, Japan), and titratable acidity (TA%; malic acid) was measured by titration with 0.1N NaOH [80]. The total sugars (TS) were determined by Bertrand’s method [80], which is based on the reducing action of sugar on an alkaline solution of tartarate complex with cupric ion. The cuprous oxide formed was dissolved in a warm acid solution of ferric alum. The ferric alum was reduced to FeSO_4_, which was titrated against standardized KMnO4; Cu equivalence was correlated with the table to obtain the amount of reducing sugar. This is based on the alkaline solution of the tartarate complex of cupric ion.

### 3.5. Fruit Weight and Yield

Harvesting took place on 7–8 July, 6–8 July and 17–20 July in 2018, 2019 and 2020, respectively. Fruit harvest was measured by the weight method considering fruit weights from each measurement tree. The weight of a single fruit was measured from a sample of 200 randomly selected fruits.

### 3.6. Statistical Analysis

The data were subject to dispersion analysis with Microsoft Office Excel 2007. The means were compared with the LSD test. To determine the relationship between the content of potassium in soil and plants, a Pearson correlation analysis was performed, the statistical significance of which was evaluated by Student’s t-criterion. The results are expressed as mean ± standard deviation for three repetitions. The significance level for all statistical analyses was 95% (*p* < 0.05).

## 4. Conclusions

The present study clarified some characteristics of the potassium regime in the loamy Haplic Luvisol under a rainfed sour cherry orchard. While the supplies of mobile and fixed potassium compounds in unfertilized soil were linked to soil mineralogy and texture, the interannual and seasonal dynamics of plant-available potash depended on the weather patterns and the uptake of potassium by trees. In 2020 (6th year after tree planting), the first signs of potassium nutrition insufficiency appeared, such as low leaf and fruit potassium status and a decrease in non-exchangeable potassium reserves in the 20–40 cm soil layer in unfertilized plots.

The annual ground fertilization with potash and nitrogen in rates from N30K40 to N120K160 led to the gradual accumulation of exchangeable potassium at depths of 0–20, 20–40 and 40–60 cm. The accumulation rate increased with the fertilizer rate increase.

When the exchangeable potassium level in the topsoil reached 200 mg/kg, the intensification of both the seasonal fluctuations in potash content and the potash leaching into the soil depths occurred in all experimental treatments. Thus, the content of exchangeable potassium exceeding the above-mentioned level was excessive, and the K+ cations were more weakly fixed by the soil-adsorbing complex.

The presented results confirm the expediency of regular fertilizer application with the rates appropriate for the conservation of potassium balance in ‘soil–tree’ systems. In the conditions of our experiment, one-time treatments with superfluous potassium rates (over 80 kg/ha) did not provide an increased stock of plant-available potash in soil but caused unreasonable losses of it due to leaching.

The evaluation of leaf and fruit potassium status and yield and fruit quality parameters correlates with the results of soil diagnostics. An increase in fertilizer rates was not essential for normal metabolic processes and did not manifest itself as an increase in potassium content in leaves and fruits or as an increase in yield.

Since our experiment was carried out in a relatively young sour cherry orchard, the augmentation of tree needs for potassium in the course of their growth and productivity increase should be considered. Also, the potassium removal rates may increase in the future, and for keeping its balance in the ‘soil–tree’ system, higher fertilization rates will be required.

## Figures and Tables

**Figure 1 plants-12-03131-f001:**
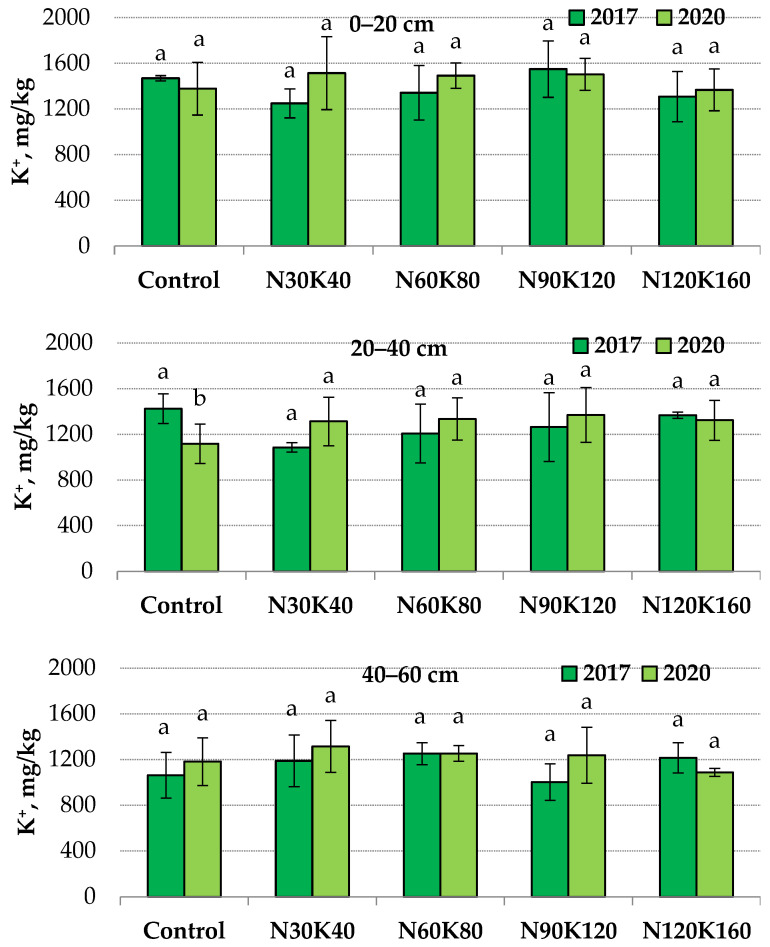
Non-exchangeable potassium content in soil in 2017 (1st year of fertilization) and 2020 (4th year of fertilization) at depths of 0–20, 20–40 and 40–60 cm. Data (mean ± standard error) followed by different letters above the bars indicate significant differences among years at *p* < 0.05 (LSD test).

**Figure 2 plants-12-03131-f002:**
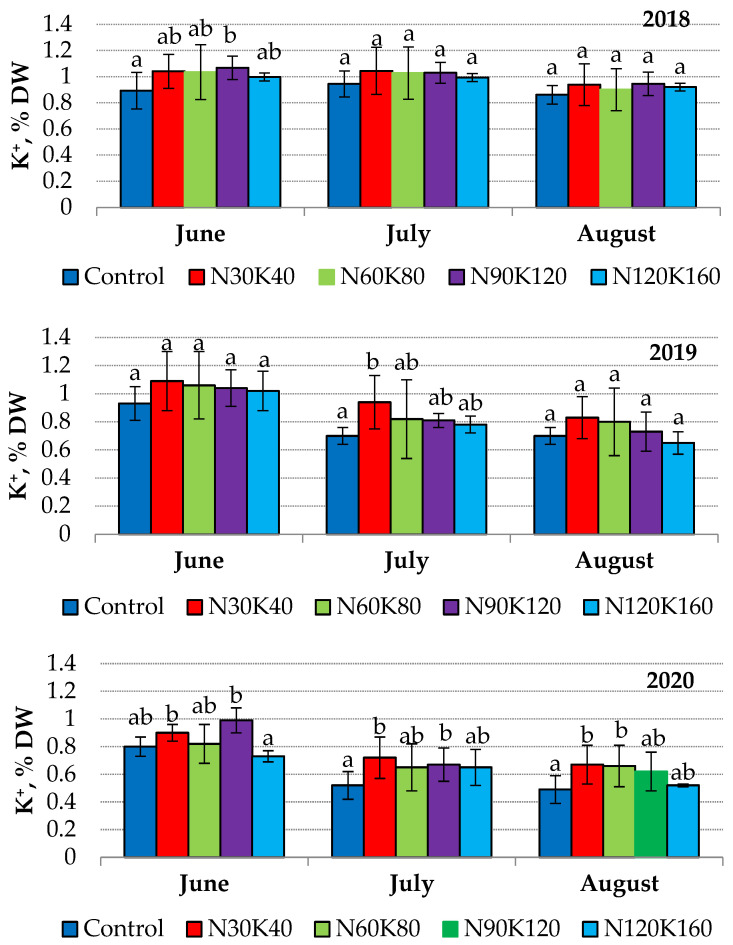
Seasonal dynamics of potassium content (% DW) in leaves of sour cherry cv. ‘Turgenevka’. Data (mean ± standard error) followed by different letters above the bars indicate significant differences among treatments at *p* < 0.05 (LSD test).

**Table 1 plants-12-03131-t001:** Interannual dynamics of soil exchangeable potassium (mg/kg) as measured at 0–20, 20–40 and 40–60 cm depths.

Treatments	Year								
	2018	2019	2020	2018	2019	2020	2018	2019	2020
	Depth 0–20 cm			Depth 20–40 cm			Depth 40–60 cm		
Control	148.9 ± 11.6 ^a^	158.5 ± 15.3 ^a^	160.6 ± 11.3 ^a^	97.0 ± 9.1 ^a^	94.1 ± 5.8 ^a^	83.3 ± 19.8 ^a^	65.8 ± 7.7 ^a^	68.6 ± 7.7 ^a^	57.9 ± 14,4 ^a^
N30K40	155.9 ± 13.7 ^a^	191.5 ± 17.9 ^a^	202.3 ± 17.3 ^b^	96.9 ± 7.0 ^a^	114.9 ± 7.2 ^a^	89.8 ± 22.6 ^a^	64.9 ± 8.8 ^a^	83.1 ± 7.5 ^a^	58.1 ± 16.3 ^a^
N60K80	219.3 ± 14.6 ^b^	212.7 ± 22.4 ^b^	214.8 ± 21.6 ^b^	132.8 ± 10.0 ^b^	119.7 ± 15.8 ^a^	124.0 ± 22.3 ^b^	68.0 ± 11.8 ^a^	80.4 ± 7.4 ^a^	81.5 ± 28.2 ^b^
N90K120	260.4 ± 35.5 ^b^	218.4 ± 18.5 ^b^	254.5 ± 24.0 ^b^	117.9 ± 13.4 ^b^	143.4 ± 22.4 ^b^	127.2 ± 38.5 ^b^	76.0 ± 10.9 ^a^	87.7 ± 16.8 ^a^	88.5 ± 19.4 ^b^
N120K160	198.6 ± 15.6 ^b^	198.3 ± 20.2 ^b^	219.3 ± 11.5 ^b^	107.2 ± 7.0 ^a^	114.4 ± 14.0 ^a^	111.4 ± 30.9 ^b^	72.3 ± 4.8 ^a^	91.4 ± 21.8 ^a^	77.5 ± 25.9 ^b^

Data are mean ± standard deviation (SD) from May to September. Values marked with different letters are statistically different within the columns at *p* ≤ 0.05 (LSD test).

**Table 2 plants-12-03131-t002:** Seasonal dynamics of soil exchangeable potassium (mg/kg) as measured at 0–20, 20–40 and 40–60 cm depths in 2018.

Treatments	Months				
	May	June	July	August	September
depth 0–20 cm					
Control	123.6 ± 2.6 ^a^	171.0 ± 7.4 ^a^	182.3 ± 13.3 ^a^	125.8 ± 4.1 ^a^	142.0 ± 20.3 ^a^
N30K40	175.9 ± 13.6 ^ab^	143.6 ± 2.8 ^a^	178.4 ± 56.9 ^a^	129.1 ± 5.0 ^a^	149.1 ± 17.0 ^a^
N60K80	228.7 ± 63.8 ^ab^	232.8 ± 19.6 ^ab^	232.5 ± 12.5 ^ab^	191.2 ± 11.7 ^ab^	211.3 ± 22.5 ^a^
N90K120	270.0 ± 62.5 ^b^	281.9 ± 83.0 ^b^	303.6 ± 80.3 ^b^	236.2 ± 35.3 ^b^	210.4 ± 29.6 ^a^
N120K160	180.6 ± 18.6 ^ab^	212.0 ± 6.5 ^ab^	227.1 ± 17.8 ^ab^	162.3 ± 15.2 ^ab^	202.0 ± 60.0 ^a^
Mean	195.8 ± 55.7 ^AB^	208.3 ± 29.0 ^AB^	225.5 ± 50.6 ^B^	168.9 ± 46.1 ^A^	182.9 ± 34.4 ^AB^
depth 20–40 cm					
Control	84.6 ± 16.1 ^a^	91.6 ± 7.1 ^a^	99.8 ± 17.7 ^a^	113.9 ± 37.4 ^a^	97.9 ± 0.5 ^a^
N30K40	96.1 ± 20.1 ^a^	93.8 ± 11.8 ^a^	93.6 ± 12.9 ^a^	81.0 ± 8.2 ^a^	114.3 ± 8.0 ^ab^
N60K80	111.4 ± 27.2 ^a^	134.2 ± 24.7 ^a^	147.6 ± 8.8 ^b^	121.6 ± 18.5 ^a^	146.9 ± 14.3 ^b^
N90K120	123.7 ± 52.9 ^a^	126.6 ± 31.8 ^a^	133.2 ± 22.8 ^ab^	114.8 ± 2.8 ^a^	91.3 ± 13.5 ^a^
N120K160	91.0 ± 4.7 ^a^	120.1 ± 9.2 ^a^	109.5 ± 15.5 ^ab^	120.4 ± 4.9 ^a^	95.0 ± 14.5 ^a^
Mean	101.2 ± 15.9 ^A^	117.0 ± 19.4 ^A^	114.1 ± 22.9 ^A^	110.3 ± 16.7 ^A^	109.1 ± 22.9 ^A^
depth 40–60 cm					
Control	46.0 ± 15.8 ^a^	70.6 ± 17.6 ^a^	87.1 ± 1.0 ^a^	53.5 ± 1.2 ^a^	71.9 ± 1.3 ^a^
N30K40	50.3 ± 9.4 ^a^	73.8 ± 6.9 ^a^	74.5 ± 6.5 ^a^	47.1 ± 8.8 ^a^	78.7 ± 32.2 ^a^
N60K80	50.6 ± 15.5 ^a^	63.8 ± 3.4 ^a^	68.4 ± 8.3 ^a^	60.4 ± 3.5 ^a^	97.0 ± 50.5 ^a^
N90K120	98.6 ± 46.3 ^b^	72.9 ± 11.7 ^a^	87.5 ± 4.2 ^a^	66.6 ± 2.1 ^a^	54.5 ± 3.0 ^a^
N120K160	60.5 ± 7.0 ^ab^	78.1 ± 5.9 ^a^	79.6 ± 3.8 ^a^	68.0 ± 3.4 ^a^	75.2 ± 14.9 ^a^
Mean	61.2 ± 21.5 ^AB^	71.8 ± 5.3 ^AB^	79.4 ± 8.2 ^B^	59.1 ± 8.8 ^A^	75.5 ± 15.2 ^AB^

Values are mean ± SD. Values marked with different lower-case letters are statistically different within the columns and values marked with different upper-case letters are statistically different within the rows at *p* ≤ 0.05 (LSD test).

**Table 3 plants-12-03131-t003:** Seasonal dynamics of soil exchangeable potassium (mg/kg) as measured at 0–20, 20–40 and 40–60 cm depths in 2019.

Treatments	Months				
	May	June	July	August	September
depth 0–20 cm					
Control	182.8 ± 6.20 ^a^	136.5 ± 38.5 ^a^	118.6 ± 22.3 ^a^	183.6 ± 22.6 ^a^	171.2 ± 21.2 ^a^
N30K40	205.1 ± 26.7 ^a^	138.8 ± 2.6 ^a^	172.0 ± 12.2 ^a^	207.0 ± 37.7 ^a^	234.5 ± 30.3 ^a^
N60K80	243.5 ± 36.7 ^a^	185.3 ± 29.9 ^a^	198.9 ± 53.9 ^a^	214.9 ± 59.8 ^a^	222.5 ± 68.3 ^a^
N90K120	244.0 ± 58.7 ^a^	212.1 ± 32.7 ^a^	182.4 ± 8.8 ^a^	230.6 ± 55.8 ^a^	222.9 ± 16.5 ^a^
N120K160	205.5 ± 34.0 ^a^	151.5 ± 36.6 ^a^	175.8 ± 10.8 ^a^	248.5 ± 60.7 ^a^	207.3 ± 4.6 ^a^
Mean	216.0 ± 26.9 ^B^	164.8 ± 32.8 ^A^	169.5 ± 30.3 ^A^	216.7 ± 24.2 ^B^	211.5 ± 24.7 ^B^
depth 20–40 cm					
Control	93.8 ± 12.9 ^a^	98.6 ± 14.3 ^a^	81.1 ± 12.7 ^a^	99.8 ± 15.7 ^a^	97.0 ± 2.80 ^a^
N30K40	113.6 ± 10.6 ^ab^	142.3 ± 3.4 ^a^	104.8 ± 9.1 ^a^	111.4 ± 1.3 ^a^	102.2 ± 11.0 ^a^
N60K80	139.1 ± 15.0 ^ab^	137.8 ± 51.8 ^a^	104.4 ± 26.2 ^a^	101.4 ± 44.1 ^a^	116.0 ± 27.7 ^a^
N90K120	176.5 ± 69.8 ^b^	164.4 ± 92.6 ^a^	112.8 ± 16.1 ^a^	143.5 ± 5.3 ^a^	119.9 ± 21.7 ^a^
N120K160	109.4 ± 9.4 ^ab^	153.8 ± 39.2 ^a^	91.5 ± 10.3 ^a^	109.4 ± 24.6 ^a^	108.0 ± 12.6 ^a^
Mean	126.5 ± 32.4 ^AB^	139.4 ± 25 ^B^	98.9 ± 12.5 ^A^	113.1 ± 17.7 ^AB^	108.6 ± 9.5 ^AB^
depth 40–60 cm					
Control	77.3 ± 20.7 ^a^	66.2 ± 18.1 ^a^	58.4 ± 12.3 ^a^	79.1 ± 17.5 ^a^	61.9 ± 17.7 ^a^
N30K40	101.9 ± 2.7 ^a^	95.2 ± 21.9 ^a^	65.0 ± 7.0 ^a^	73.6 ± 3.7 ^a^	79.7 ± 4.1 ^a^
N60K80	87.5 ± 4.8 ^a^	79.9 ± 27.5 ^a^	81.3 ± 14.7 ^a^	78.5 ± 15.4 ^a^	74.9 ± 20.3 ^a^
N90K120	83.6 ± 38.6 ^a^	115.3 ± 74.1 ^a^	71.8 ± 4.8 ^a^	81.4 ± 16.4 ^a^	86.2 ± 17.0 ^a^
N120K160	92.9 ± 4.7 ^a^	129.7 ± 71.1 ^a^	70.7 ± 6.5 ^a^	87.4 ± 25.8 ^a^	76.3 ± 3.0 ^a^
Mean	88.6 ± 9.3 ^AB^	97.3 ± 25.7 ^B^	69.4 ± 8.5 ^A^	80.0 ± 5.0 ^AB^	75.8 ± 8.9 ^AB^

Values are mean ± SD. Values marked with different lower-case letters are statistically different within the columns and values marked with different upper-case letters are statistically different within the rows at *p* ≤ 0.05 (LSD test).

**Table 4 plants-12-03131-t004:** Seasonal dynamics of soil exchangeable potassium (mg/kg) as measured at 0–20, 20–40 and 40–60 cm depths in 2020.

Treatments	Months				
	May	June	July	August	September
depth 0–20 cm					
Control	172.0 ± 6.5 ^a^	142.0 ± 5.6 ^a^	187.7 ± 35.1 ^a^	153.2 ± 7.1 ^a^	148.1 ± 16.4 ^a^
N30K40	218.8 ± 3.8 ^a^	229.1 ± 49.8 ^b^	212.0 ± 44.0 ^a^	177.7 ± 26.5 ^a^	173.7 ± 23.1 ^a^
N60K80	236.0 ± 33.7 ^a^	193.8 ± 39.1 ^a^	256.3 ± 31.4 ^a^	179.7 ± 54.5 ^a^	208.4 ± 42.9 ^a^
N90K120	254.8 ± 53.4 ^b^	270.2 ± 65.6 ^b^	250.6 ± 2.4 ^a^	236.6 ± 9.8 ^b^	260.3 ± 55.6 ^b^
N120K160	232.6 ± 14.1 ^a^	208.8 ± 22.9 ^a^	253.6 ± 9.7 ^a^	193.9 ± 14.2 ^a^	207.8 ± 18.1 ^a^
Mean	222.8 ± 31.2 ^B^	208.8 ± 47.1 ^AB^	231.7 ± 30.7 ^B^	188.2 ± 30.8 ^A^	199.6 ± 42.3 ^AB^
depth 20–40 cm					
Control	108.5 ± 11.3 ^a^	90.3 ± 8.7 ^a^	83.4 ± 1.9 ^a^	77.4 ± 14.3 ^a^	56.9 ± 7.3 ^a^
N30K40	124.8 ± 7.4 ^a^	96.0 ± 12.0 ^a^	87.6 ± 5.2 ^a^	77.1 ± 4.2 ^a^	63.7 ± 5.4 ^a^
N60K80	133.1 ± 22.7 ^a^	113.8 ± 28.7 ^a^	144.6 ± 44.0 ^b^	111.2 ± 25.3 ^a^	117.1 ± 21.5 ^b^
N90K120	125.2 ± 6.8 ^a^	119.4 ± 14.3 ^a^	112.2 ± 7.7 ^a^	108.5 ± 51.5 ^a^	96.0 ± 10.6 ^a^
N120K160	123.9 ± 13.9 ^a^	117.1 ± 44.5 ^a^	132.1 ± 37.9 ^a^	97.8 ± 8.3 ^a^	85.9 ± 15.1 ^a^
Mean	123.1 ± 8.9 ^B^	107.3 ± 13.2 ^B^	112.0 ± 26.8 ^B^	94.4 ± 16.4 ^AB^	83.9 ± 24.4 ^A^
depth 40–60 cm					
Control	75.4 ± 6.9 ^a^	66.9 ± 11.1 ^a^	53.4 ± 2.3 ^a^	53.5 ± 2.4 ^a^	40.1 ± 9.2 ^a^
N30K40	72.0 ± 7.1 ^a^	57.1 ± 1.1 ^a^	66.8 ± 14.5 ^a^	35.8 ± 4.9 ^a^	58.9 ± 17.9 ^a^
N60K80	103.9 ± 9.5 ^a^	89.8 ± 44.5 ^a^	93.9 ± 10.7 ^b^	63.6 ± 16.2 ^a^	56.4 ± 15.7 ^a^
N90K120	89.9 ± 16.3 ^a^	88.5 ± 5.8 ^a^	97.1 ± 13.9 ^b^	95.4 ± 16.6 ^b^	58.6 ± 7.0 ^a^
N120K160	83.8 ± 5.0 ^a^	103.9 ± 41.4 ^b^	84.2 ± 11.5 ^a^	61.1 ± 8.4 ^a^	54.2 ± 3.1 ^a^
Mean	85.0 ± 12.7 ^B^	81.2 ± 18.9 ^B^	79.5 ± 18.6 ^B^	61.7 ± 21.7 ^A^	53.7 ± 7.8 ^A^

Values are mean ± SD. Values marked with different lower-case letters are statistically different within the columns values marked with different upper-case letters are statistically different within the rows at *p* ≤ 0.05 (LSD test).

**Table 5 plants-12-03131-t005:** Interannual dynamics of soil water-soluble potassium (mg/kg) as measured at 0–20, 20–40 and 40–60 cm depths.

Treatments	Year								
	2018	2019	2020	2018	2019	2020	2018	2019	2020
	Depth 0–20 cm			Depth 20–40 cm			Depth 40–60 cm		
Control	15.3 ± 4.5 ^a^	15.3 ± 4.2 ^a^	14.8 ± 2.8 ^a^	10.5 ± 2.8 ^a^	8.8 ± 2.1 ^a^	8.2 ± 3.4 ^a^	8.3 ± 3.7 ^a^	6.1 ± 1.5 ^a^	6.4 ± 0.9 ^a^
N30K40	17.2 ± 5.0 ^a^	27.1 ± 11.9 ^b^	22.8 ± 11.2 ^a^	12.1 ± 3.8 ^a^	11.0 ± 5.9 ^a^	8.8 ± 1.6 ^a^	12.7 ± 2.8 ^a^	8.8 ± 1.8 ^a^	6.2 ± 0.8 ^a^
N60K80	25.2 ± 10.5 ^a^	28.0 ± 12.4 ^b^	30.3 ± 9.8 ^b^	15.2 ± 4.2 ^b^	11.1 ± 3.7 ^a^	12.8 ± 4.1 ^b^	8.3 ± 3.6 ^a^	7.8 ± 2.7 ^a^	8.3 ± 2.3 ^b^
N90K120	41.4 ± 22.1 ^b^	28.2 ± 11.4 ^b^	42.0 ± 14.9 ^b^	15.0 ± 6.5 ^b^	15.8 ± 6.9 ^a^	11.1 ± 6.2 ^b^	10.7 ± 4.1 ^a^	10.4 ± 3.1 ^a^	8.3 ± 2.0 ^b^
N120K160	35.8 ± 4.5 ^b^	29.0 ± 14.3 ^b^	33.7 ± 9.2 ^b^	13.0 ± 2.7 ^a^	12.8 ± 1.9 ^a^	11.8 ± 4.0 ^b^	9.7 ± 3.7 ^a^	10.4 ± 1.8 ^a^	7.5 ± 1.7 ^ab^

Data are mean ± SD over May to September. Values marked with different letters are statistically different within the columns at *p* ≤ 0.05 (LSD test).

**Table 6 plants-12-03131-t006:** Seasonal dynamics of soil water-soluble potassium (mg/kg) as measured at 0–20, 20–40 and 40–60 cm depths in 2018.

Treatments	Months				
	May	June	July	August	September
depth 0–20 cm					
Control	12.6 ± 1.9 ^a^	18.6 ± 0.5 ^a^	20.5 ± 0.4 ^a^	12.9 ± 6.2 ^a^	12.1 ± 3.3 ^a^
N30K40	20.7 ± 3.6 ^a^	16.6 ± 3.4 ^a^	20.9 ± 6.7 ^a^	12.5 ± 4.9 ^a^	15.2 ± 1.0 ^ab^
N60K80	38.8 ± 15.7 ^b^	37.5 ± 4.6 ^a^	28.6 ± 2.8 ^a^	19.0 ± 0.9 ^a^	24.4 ± 4.6 ^ab^
N90K120	68.4 ± 17.8 ^c^	33.7 ± 11.1 ^a^	50.1 ± 27.9 ^b^	26.5 ± 5.8 ^a^	28.7 ± 8.2 ^ab^
N120K160	41.5 ± 8.2 ^b^	42.4 ± 20.0 ^b^	35.7 ± 9.6 ^ab^	21.4 ± 4.1 ^a^	37.9 ± 18.0 ^b^
Mean	31.9 ± 21.6 ^B^	29.8 ± 11.5 ^B^	31.2 ± 12.3 ^B^	18.4 ± 5.8 ^A^	23.7 ± 10.4 ^AB^
depth 20–40 cm					
Control	11.0 ± 4.0 ^a^	13.6 ± 2.1 ^a^	10.4 ± 0.9 ^a^	9.2 ± 3.6 ^a^	8.2 ± 2.3 ^a^
N30K40	10.9 ± 1.9 ^a^	17.5 ± 3.0 ^a^	10.6 ± 1.5 ^a^	7.4 ± 1.3 ^a^	14.0 ± 3.1 ^ab^
N60K80	14.4 ± 1.1 ^a^	15.5 ± 4.7 ^a^	17.0 ± 5.0 ^a^	10.4 ± 3.8 ^a^	18.5 ± 2.3 ^b^
N90K120	16.1 ± 4.5 ^a^	19.5 ± 11.3 ^a^	17.1 ± 6.6 ^a^	11.0 ± 4.1 ^a^	11.2 ± 1.3 ^ab^
N120K160	13.0 ± 4.0 ^a^	15.7 ± 3.2 ^a^	10.9 ± 0.9 ^a^	10.6 ± 1.5 ^a^	14.6 ± 1.2 ^ab^
Mean	13.1 ± 2.2 ^AB^	16.4 ± 2.2 ^B^	13.2 ± 3.5 ^AB^	9.7 ± 1.5 ^A^	13.3 ± 3.9 ^AB^
depth 40–60 cm					
Control	8.6 ± 1.7 ^a^	12.0 ± 6.4 ^a^	7.9 ± 2.3 ^a^	5.2 ± 2.0 ^a^	7.6 ± 1.8 ^a^
N30K40	7.5 ± 0.4 ^a^	9.5 ± 1.2 ^a^	10.5 ± 5.7 ^a^	4.2 ± 0.3 ^a^	9.4 ± 0.5 ^a^
N60K80	7.6 ± 1.1 ^a^	10.2 ± 1.9 ^a^	6.8 ± 2.0 ^a^	6.7 ± 1.9 ^a^	12.0 ± 5.0 ^a^
N90K120	14.0 ± 5.2 ^a^	12.8 ± 4.5 ^a^	11.5 ± 3.4 ^a^	7.6 ± 1.9 ^a^	7.6 ± 1.6 ^a^
N120K160	9.5 ± 0.4 ^a^	12.2 ± 7.4 ^a^	7.0 ± 1.5 ^a^	9.4 ± 5.5 ^a^	10.6 ± 1.4 ^a^
Mean	13.9 ± 2.7 ^B^	11.3 ± 1.4 ^B^	8.4 ± 2.6 ^AB^	6.6 ± 2.0 ^A^	9.4 ± 1.9 ^AB^

Values are mean ± SD. Values marked with different lower-case letters are statistically different within the columns and values marked with different upper-case letters are statistically different within the rows at *p* ≤ 0.05 (LSD test).

**Table 7 plants-12-03131-t007:** Seasonal dynamics of soil water-soluble potassium (mg/kg) as measured at 0–20, 20–40 and 40–60 cm depths in 2019.

Treatments	Months				
	May	June	July	August	September
depth 0–20 cm					
Control	14.1 ± 0.6 ^a^	10.5 ± 3.6 ^a^	14.8 ± 3.3 ^a^	20.2 ± 3.5 ^a^	17.0 ± 2.4 ^a^
N30K40	32.9 ± 12.0 ^a^	11.3 ± 0.9 ^a^	27.0 ± 3.5 ^a^	28.2 ± 9.4 ^ab^	35.9 ± 6.5 ^a^
N60K80	32.4 ± 14.3 ^a^	18.8 ± 3.3 ^a^	27.9 ± 8.4 ^a^	26.9 ± 10.2 ^ab^	34.0 ± 18.8 ^a^
N90K120	32.4 ± 14.6 ^a^	24.1 ± 7.5 ^a^	22.8 ± 2.9 ^a^	36.8 ± 15.9 ^ab^	25.0 ± 4.9 ^a^
N120K160	28.1 ± 5.1 ^a^	18.2 ± 8.3 ^a^	27.9 ± 6.0 ^a^	46.6 ± 22.4 ^b^	24.3 ± 1.7 ^a^
Mean	28.0 ± 7.8 ^B^	16.6 ± 5.7 ^A^	24.1 ± 5.6 ^AB^	31.7 ± 10.2 ^B^	27.2 ± 7.7 ^B^
depth 20–40 cm					
Control	7.1 ± 1.3 ^a^	7.1 ± 0.5 ^a^	9.1 ± 2.9 ^a^	10.6 ± 3.0 ^a^	10.2 ± 2.2 ^a^
N30K40	9.9 ± 1.0 ^a^	12.4 ± 2.2 ^ab^	12.9 ± 3.5 ^a^	10.4 ± 1.5 ^a^	9.4 ± 2.4 ^a^
N60K80	12.5 ± 1.9 ^a^	12.8 ± 7.0 ^ab^	10.0 ± 3.7 ^a^	10.7 ± 2.6 ^a^	9.5 ± 2.0 ^a^
N90K120	18.6 ± 11.4 ^a^	23.6 ± 21.3 ^b^	11.2 ± 2.3 ^a^	15.5 ± 3.2 ^a^	10.0 ± 1.1 ^a^
N120K160	8.8 ± 1.5 ^a^	23.8 ± 16.2 ^b^	10.5 ± 2.1 ^a^	11.2 ± 2.9 ^a^	9.5 ± 2.0 ^a^
Mean	13.0 ± 4.5 ^A^	15.9 ± 7.4 ^A^	10.7 ± 1.4 ^A^	11.7 ± 2.2 ^A^	9.7 ± 0.4 ^A^
depth 40–60 cm					
Control	5.1 ± 0.7 ^a^	5.3 ± 1.2 ^a^	6.5 ± 2.2 ^a^	6.9 ± 2.3 ^a^	6.5 ± 0.4 ^a^
N30K40	9.9 ± 1.3 ^a^	8.9 ± 2.8 ^ab^	9.8 ± 2.0 ^a^	7.8 ± 0.9 ^a^	7.4 ± 1.1 ^a^
N60K80	6.4 ± 0.7 ^a^	6.0 ± 2.1 ^a^	11.5 ± 5.4 ^a^	8.3 ± 2.4 ^a^	6.8 ± 0.9 ^a^
N90K120	6.9 ± 3.1 ^a^	17.9 ± 7.4 ^ab^	11.4 ± 4.5 ^a^	7.6 ± 2.0 ^a^	8.3 ± 2.2 ^a^
N120K160	6.8 ± 0.9 ^a^	20.8 ± 8.9 ^b^	7.9 ± 2.1 ^a^	9.8 ± 2.6 ^a^	6.8 ± 1.2 ^a^
Mean	7.0 ± 1.8 ^A^	11.8 ± 7.1 ^A^	9.4 ± 2.2 ^A^	8.1 ± 1.1 ^A^	7.2 ± 0.7 ^A^

Values are mean ± SD. Values marked with different lower-case letters are statistically different within the columns and values marked with different upper-case letters are statistically different within the rows at *p* ≤ 0.05 (LSD test).

**Table 8 plants-12-03131-t008:** Seasonal dynamics of soil water-soluble potassium (mg/kg) as measured at 0–20, 20–40 and 40–60 cm depths in 2020.

Treatments	Months				
	May	June	July	August	September
depth 0–20 cm					
Control	15.2 ± 0.5 ^a^	13.4 ± 1.9 ^a^	17.6 ± 4.5 ^a^	13.1 ± 3.0 ^a^	14.9 ± 1.0 ^a^
N30K40	21.9 ± 0.5 ^a^	33.6 ± 15.2 ^ab^	22.7 ± 7.5 ^a^	16.2 ± 4.1 ^a^	19.7 ± 7.8 ^a^
N60K80	36.5 ± 4.6 ^ab^	26.4 ± 3.7 ^a^	36.7 ± 9.9 ^a^	20.9 ± 7.4 ^a^	31.0 ± 12.1 ^ab^
N90K120	49.7 ± 20.0 ^b^	57.9 ± 22.9 ^b^	30.0 ± 3.2 ^a^	29.5 ± 2.3 ^a^	43.2 ± 24.5 ^b^
N120K160	38.5 ± 6.6 ^ab^	25.9 ± 3.2 ^a^	40.7 ± 1.7 ^a^	28.9 ± 2.2 ^a^	34.5 ± 15.6 ^ab^
Mean	32.4 ± 13.8 ^A^	31.4 ± 16.5 ^A^	29.5 ± 9.6 ^A^	21.7 ± 7.4 ^A^	24.2 ± 11.4 ^A^
depth 20–40 cm					
Control	10.5 ± 1.1 ^a^	8.7 ± 2.0 ^a^	7.4 ± 1.5 ^a^	7.2 ± 2.5 ^a^	7.0 ± 0.4 ^a^
N30K40	11.1 ± 1.1 ^a^	9.7 ± 1.0 ^a^	7.6 ± 1.0 ^a^	7.4 ± 0.6 ^a^	8.1 ± 0.8 ^ab^
N60K80	12.4 ± 1.4 ^a^	11.6 ± 1.8 ^a^	16.4 ± 7.4 ^b^	10.4 ± 3.0 ^a^	13.4 ± 2.8 ^b^
N90K120	11.4 ± 1.3 ^a^	12.1 ± 2.0 ^a^	10.0 ± 0.9 ^ab^	12.7 ± 3.5 ^a^	9.6 ± 0.9 ^ab^
N120K160	11.7 ± 2.6 ^a^	11.6 ± 3.6 ^a^	15.7 ± 6.7 ^b^	10.5 ± 0.3 ^a^	9.7 ± 2.0 ^ab^
Mean	11.4 ± 0.7 ^A^	10.7 ± 1.5 ^A^	11.4 ± 4.4 ^A^	9.6 ± 2.3 ^A^	9.6 ± 2.4 ^A^
depth 40–60 cm					
Control	7.1 ± 0.7 ^a^	6.7 ± 0.5 ^a^	5.6 ± 1.0 ^a^	5.5 ± 0.7 ^a^	7.0 ± 0.4 ^a^
N30K40	6.2 ± 1.0 ^a^	6.2 ± 0.5 ^a^	6.2 ± 0.3 ^a^	5.4 ± 0.3 ^a^	7.2 ± 1.5 ^a^
N60K80	10.4 ± 2.8 ^b^	8.7 ± 2.9 ^a^	8.5 ± 0.6 ^a^	6.6 ± 1.6 ^a^	7.4 ± 1.3 ^a^
N90K120	9.2 ± 2.9 ^ab^	9.5 ± 0.4 ^a^	7.7 ± 0.5 ^a^	8.2 ± 2.0 ^a^	6.7 ± 1.5 ^a^
N120K160	6.6 ± 0.8 ^a^	9.4 ± 3.1 ^a^	7.6 ± 1.9 ^a^	6.9 ± 0.5 ^a^	6.9 ± 0.3 ^a^
Mean	7.9 ± 1.8 ^AB^	8.1 ± 1.5 ^B^	7.1 ± 1.2 ^AB^	6.5 ± 1.1 ^A^	7.0 ± 0.3 ^AB^

Values are mean ± SD. Values marked with different lower-case letters are statistically different within the columns and values marked with different upper-case letters are statistically different within the rows at *p* ≤ 0.05 (LSD test).

**Table 9 plants-12-03131-t009:** Fruit weight and fruit yield of cv. ‘Turgenevka’ sour cherry trees.

Parameter	Year	Treatments				
		Control	N30K40	N60K80	N90K120	N120K160
fruit weight (g)	2018	5.15 ± 0.09 ^a^	4.98 ± 0.32 ^a^	5.07 ± 0.08 ^a^	4.79 ± 0.06 ^a^	5.31 ± 0.22 ^a^
	2019	4.38 ± 0.28 ^a^	4.49 ± 0.17 ^a^	4.61 ± 0.04 ^a^	4.58 ± 0.11 ^a^	4.53 ± 0.37 ^a^
	2020	5.02 ± 0.20 ^a^	5.00 ± 0.13 ^a^	4.92 ± 0.22 ^a^	4.93 ± 0.24 ^a^	5.13 ± 0.12 ^a^
fruit yield(kg/tree)	2018	4.38 ± 0.28 ^a^	4.26 ± 0.69 ^a^	3.90 ± 1.10 ^a^	5.12 ± 1.40 ^a^	5.38 ± 0.57 ^a^
	2019	8.24 ± 1.41 ^a^	8.46 ± 3.11 ^a^	8.67 ± 1.91 ^a^	7.01 ± 1.52 ^a^	8.33 ± 2.13 ^a^
	2020	5.97 ± 1.93 ^a^	7.77 ± 1.67 ^a^	9.23 ± 2.37 ^b^	8.88 ± 0.78 ^a^	10.11 ± 3.50 ^b^

Values are mean ± SD. Values marked with different letters are statistically different within the rows at *p* ≤ 0.05 (LSD test).

**Table 10 plants-12-03131-t010:** Chemical composition of cv. ‘Turgenevka’ sour cherry fruits.

Year(Factor A)	Treatments (Factor B)					Mean A
	Control	N30K40	N60K80	N90K120	N120K160	
	Potassium content (mg/100 g FW)					
2018	172.03 ± 4.56 ^a^	175.76 ± 10.50 ^a^	180.55 ± 6.64 ^a^	193.01 ± 11.91 ^b^	172.05 ± 8.09 ^a^	178.70 ± 4.30 ^B^
2019	189.35 ± 5.70 ^ab^	190.01 ± 8.56 ^b^	181.87 ± 15.05 ^ab^	174.06 ± 14.73 ^a^	178.78 ± 8.30 ^ab^	182.80 ± 5.13 ^B^
2020	149.32 ± 7.77 ^a^	167.95 ± 10.98 ^b^	165.34 ± 16.00 ^b^	173.33 ± 9.21 ^b^	149.73 ± 8.14 ^a^	161.13 ± 5.75 ^A^
Mean B	170.23 ± 8.46 ^ab^	177.91 ± 7.48 ^b^	175.92 ± 8.14 ^b^	180.13 ± 8.55 ^b^	166.85 ± 7.53 ^a^	
	Total soluble solids (TSS),%					
2018	15.03 ± 0.81 ^a^	16.00 ± 1.01 ^a^	15.43 ± 1.36 ^a^	15.96 ± 0.93 ^a^	15.30 ± 0.53 ^a^	15.54 ± 0.9 ^A^
2019	16.00 ± 3.7 ^a^	15.16 ± 0.29 ^a^	15.56 ± 0.84 ^a^	15.83 ± 0.93 ^a^	16.06 ± 0.06 ^a^	15.72 ± 1.52 ^A^
2020	16.03 ± 0.81 ^a^	17.76 ± 0.76 ^a^	17.93 ± 1.41 ^a^	16.83 ± 0.51 ^a^	16.50 ± 0.10 ^a^	17.01 ± 1.02 ^B^
Mean B	15.68 ± 2.00 ^a^	16.30 ± 1.32 ^a^	16.30 ± 1.62 ^a^	16.20 ± 0.85 ^a^	15.95 ± 0.59 ^a^	
	Titratable acidity (TA),%					
2018	1.64 ± 0.25 ^a^	1.50 ± 0.03 ^a^	1.56 ± 0.19 ^a^	1.55 ± 0.04 ^a^	1.60 ± 0.14 ^a^	1.57 ± 0.14 ^A^
2019	1.86 ± 0.25 ^a^	1.66 ± 0.19 ^a^	1.60 ± 0.06 ^a^	1.65 ± 0.16 ^a^	1.63 ± 0.09 ^a^	1.68 ± 0.17 ^A^
2020	1.99 ± 0.16 ^a^	2.16 ± 0.12 ^ab^	2.09 ± 0.18 ^ab^	2.35 ± 0.28 ^b^	2.08 ± 0.22 ^ab^	2.13 ± 0.21 ^B^
Mean B	1.83 ± 0.25 ^a^	1.77 ± 0.32 ^a^	1.75 ± 0.29 ^a^	1.85 ± 0.41 ^a^	1.77 ± 0.27 ^a^	
	Total sugars(TS),%					
2018	10.93 ± 0.47 ^a^	10.71 ± 0.46 ^a^	10.89 ± 0.48 ^a^	10.30 ± 0.35 ^a^	10.90 ± 0.77 ^a^	10.75 ± 0.51 ^B^
2019	11.05 ± 0.10 ^b^	9.28 ± 0.38 ^a^	9.35 ± 0.17 ^a^	9.76 ± 1.10 ^ab^	9.80 ± 0.40 ^ab^	9.84 ± 0.81 ^A^
2020	11.49 ± 0.67 ^a^	12.51 ± 0.48 ^a^	13.85 ± 2.44 ^b^	11.60 ± 0.66 ^a^	11.81 ± 0.69 ^a^	12.25 ± 1.38 ^C^
Mean B	11.15 ± 0.49 ^ab^	10.83 ± 1.45 ^ab^	11.36 ± 2.34 ^b^	10.55 ± 1.08 ^a^	10.83 ± 1.03 ^ab^	

Two-way analysis of variance test was conducted, and the data are mean ± SD. Values marked with different lower-case letters are statistically different within the rows and values marked with different upper-case letters are statistically different within the columns at *p* ≤ 0.05 (LSD test).

**Table 11 plants-12-03131-t011:** Correlation between fruit chemical composition and potassium status of fruit and leaves (*n* =45).

Chemical Composition	Potassium Content		
	Fruit	June Leaves	July Leaves
TSS	−0.24	−0.41 **	−0.51 ***
TA	−0.27	−0.16	−0.42 **
TS	−0.37 *	−0.51 ***	−0.42 **

*** Statistically significant at *p* ≤ 0.001; ** statistically significant at *p* ≤ 0.01; * statistically significant at *p* ≤ 0.05.

**Table 12 plants-12-03131-t012:** Correlation between the potassium levels in soil (exchangeable and water-soluble potassium content) and potassium levels in leaves and fruits (*n* =45).

	Leaf Potassium	Soil Exchangeable Potassium			Soil Water-Soluble Potassium		
		Soil Layers					
		0–20 cm	20–40 cm	40–60 cm	0–20 cm	20–40 cm	40–60 cm
Fruit potassium	0.65*	0.08	0.24	0.03	0.20	0.21	0.22
Leaf potassium		0.23	0.41 *	0.04	0.23	0.42 *	0.21

* Statistically significant at *p* ≤ 0.05.

**Table 13 plants-12-03131-t013:** Initial soil properties.

Soil Characteristics	Soil Depth, cm		
	0–20	20–40	40–60
pH_KCl_	5.8 ± 0.11	5.7 ± 0.07	5.7 ± 0.11
Organic matter, %	2.8 ± 0.10	2.6 ± 0.05	2.0 ± 0.08
Alkali-hydrolysable N ^a^, mg/kg	108 ± 1.6	98 ± 3.9	76 ± 17.1
Available P ^b^, mg/kg	383 ± 9.4	308 ± 16.2	118 ± 15.9
Exchangeable cations			
K ^c^, mg/kg	122 ± 7.8	88 ± 8.1	62 ± 14.8
Ca ^d^, mg/kg	3000 ± 24	3100 ± 20	2940 ± 48
Mg ^e^, mg/kg	530 ± 10	528 ± 12	576 ± 12

Numbers are mean values of five sampling locations ± SD: ^a^ determined by hydrolysis 1.0 M NaOH; ^b,c^ 0.2 M HCl extraction; ^d,e^ EDTA-Na2.

**Table 14 plants-12-03131-t014:** Average monthly air temperatures (°C) during the growing season in 2018, 2019 and 2020.

Month	Year												40-Year Average Values *
	2018				2019				2020				
	Decade				Decade				Decade				
	I	II	III	X	I	II	III	X	I	II	III	X	
	Air Temperature (°C)												
May	18.9	14.8	15.7	16.4	13.5	14.4	18.4	15.6	12.8	10.7	11.2	11.3	13.0
June	14.2	18.4	19.8	17.0	20.2	20.8	20.4	20.5	16.8	22.0	21.0	19.9	16.9
July	17.7	20.9	21.0	19.9	16.6	16.0	19.3	17.4	22.0	17.9	19.0	19.6	18.5
August	19.9	18.8	16.7	18.4	14.8	19.0	17.6	17.1	18.8	17.7	18.1	18.2	17.1
September	18.9	15.7	10.1	14.9	17.8	13.4	6.2	12.5	18.9	13.3	13.4	15.2	11.7
X values V–IX				17.3				16.6				16.8	

* The long-term average (40-year average, i.e., 1961–2000 period).

**Table 15 plants-12-03131-t015:** Monthly sums (millimeter) of precipitation during the growing season in 2018, 2019 and 2020.

Month	Year												40-Year Average Values *
	2018				2019				2020				
	Decade				Decade				Decade				
	I	II	III	Ʃ	I	II	III	Ʃ	I	II	III	Ʃ	
	Precipitation (mm)												
May	18.9	5.0	7.5	31.4	59.6	14.4	11.0	85.0	11.6	1.1	46.4	59.1	36.4
June	0.3	10.2	7.7	18.2	0.2	0.4	20.1	20.7	24.0	1.2	21.2	46.4	65.1
July	0.4	77.5	42.0	119.9	26.9	8.9	14.0	49.8	19.8	86.0	5.8	111.6	88.0
August	0.0	3.7	7.5	11.2	10.7	44.0	0.0	54.7	7.9	13.3	4.8	26.0	65.7
September	3.2	29.8	9.5	42.5	12.6	17.4	20.2	50.2	10.0	8.9	4.6	23.5	43.2
Ʃ values V–IX				226.2				260.4				266.6	

* The long-term average (40-year average, i.e., 1961–2000 period).

## Data Availability

In this study, the sour cherry cultivar ‘Turgenevka’ was used from the bioresource collection of Russian Research Institute of Fruit Crop Breeding (VNIISPK). Information about this cultivar is available on the website of the Russian Research Institute of Fruit Crop Breeding (VNIISPK).

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
