# Peer review of "The Features of Potassium Dynamics in ‘Soil–Plant’ System of Sour Cherry Orchard"

_plants, 2023, doi:10.3390/plants12173131_

Round 1

Reviewer 1 Report

The present manucript evaluates the interannual and seasonal dynamics of different potassium compounds in the soil and the potassium status of sour cherry trees affected by application of nitrogen and potash fertilizers.

Lines 23, 559: kg/ga? 

Tables 1-3 should be put in the Materials and methods.

Tables: replace commas with points in the mean values. Add also SD values in Tables 4, 5, 6, 7, 8, 9.

Replace Figures 1, 2 with tables since it is not possible to identify significant differences between means with the LSD criterion.

Revise the legend of Table 5 as follows "... 0-20, 20-40 and 40-60 cm.".

Figures 3, 4: indicate what the Latin letters mean.

Line 301: define the indicator values and how 1.3 was calculated.

Table 7: explain Factor B in the first row of the Table.

How the applied treatments were defined. What is the common fertilization practice in the growing area? Is potassium usually applied only once throughout the growing period?

Line 507: define DAFB when first cited in the text.

Line 557-561: this conclusion is based on the sole application of potassium at the beginning of the growing season.

What about the effect of N. The authors applied different combinations of N and P, so the differences in N availability may also have an effect.

Reviewer 2 Report

Dear authors,

The research gives important information of potassium levels in soil and sour cherry plant and its relationship. However, the English of the manuscript is very poor and must be revised by the native English language speaker or professional because some part of the manuscript cannot be fully understood before improving the language and soundness.

The section Material and methods seems very much shortened and some methods are very poorly described. Therefore, you must provide much more detailed descripiton of methods used especially when describing the trial design itself (repetition, design, sampling was performed in each repetition or...?). Generally, some parts are missing or have not been described in detail enough to understand and follow the results. Also, it is not clear how the correlation analysis was performed (n=?). Somewhere, you used the abbreviation LCD - did you mean LSD?

In the Results and disscusion section, you have clearly decribed the results. However, the Figuers contains informations that are not described nor specified in the Figure titles. Same for tables. In both cases, the titles must provide enough information without the need to check the description of the table/figure in the text. In some cases, you mark values with different letters to point the differences but in some columns you mark value with, for example, b and c. However, there is no value marked with c!! - please clarify this situation!

The Disscusion part must be improved in the way to explain obtained results more clearly by comparing them to the results of exisiting and similar research - please enrich your disscusion by adding more key references related to the each subsection of the Results. 

Section Conclusions must be improved to give exact reccomendations based on the results obtained as this is primarily investigation for improving the production technology and fertilizer application and dosage.

As stated before, the English of the manuscript is very poor and must be revised by the native English language speaker or professional because some part of the manuscript cannot be fully understood before improving the language and soundness. In this form, the manuscript cannot be published.

Round 2

Reviewer 1 Report

The authros addressed all my comments. Therefore, I suggest the acceptance of the manuscript in its present form.

Author Response

The authors thank the reviewer for usefull comments.

Reviewer 2 Report

Dear authors,

You have sufficiently improved the manuscript. However, there are still some issues that needs to be addressed. First, you have indicated n=3 in the tables where n represents the number of repetitions which is not common and I strongly recommend deletion of indication the n value where it refers to number of repetitions. Secondly, titles of the tables 11 and 12 must be corrected as follows:

Table 11. Pearson Correlation coefficient between fruit chemical composition and potassium status of fruit and leaves (n=45).

Table 12. Pearson Correlation coefficient among  between the potassium levels in soil potassium parameters (exchangeable and water-soluble potassium content) and potassium levels in leaves and fruits (n=45).

Concerning the English language, it has been improved. However, there are still minor grammarly and style mistakes that needs to be addressed. I will give an example based on the last sentence in the Conclusion section.

"Since our experiment carried out in relatively young sour cherry orchard it should be considered the augmentation of tree needs infor potassium in course of its growth and increasing of productivity increase. Also, the potassium removal rates may increase in the future and for keeping/ensuring and its the potassium balance in ‘soil-trees’ system a higher fertilization rates doses will require the greater will be required."
